

# Prevalence of risk factors associated with rupture of abdominal aortic aneurysm (AAA): a single center retrospective study

Sudong Liu[1,2], Caifu Long[3], Yuanjia Hong[3], Xiaodong Gu[1,2], Ruiqiang Weng[1,2] and Zhixiong Zhong[2,4]

[1] Research Experimental Center, Meizhou People's Hospital (Huangtang Hospital), Meizhou, China
[2] Guangdong Engineering Technology Research Center of Molecular Diagnostics for Cardiovascular Diseases, Meizhou, China
[3] Meizhou Clinical Medical School, Guangdong Medical University, Meizhou, China
[4] Center for Cardiovascular Diseases, Meizhou People's Hospital (Huangtang Hospital), Meizhou, China

Corresponding author
Zhixiong Zhong,
zhongzhixiong@mzrmyy.com

## ABSTRACT

**Background:** Abdominal aortic aneurysm (AAA) is a severe cardiovascular disease. The mortality rate for an AAA rupture is very high. Understanding the risk factors for AAA rupture would help AAA management, but little is known about these risk factors in the Chinese population.

**Methods:** This retrospective study included patients that were diagnosed with AAA during the last 5 years in a large national hospital in southern China. AAA patients were divided into a rupture and non-rupture group. Clinical data were extracted from the hospital medical record system. Clinical features were compared between the rupture and non-rupture groups. The associations between potential risk factors and rupture risk were evaluated using a multivariate logistic regression analysis.

**Results:** A total of 337 AAA patients were included for analysis in the present study. AAA diameter was significantly larger, and high-sensitivity C-reactive protein (hs-CRP) and serum creatinine levels were both significantly higher in AAA rupture patients. High-density lipoprotein cholesterol (HDL-C), low-density lipoprotein cholesterol (LDL-C) and total cholesterol (TC) levels were significantly lower in AAA rupture patients. After adjustment, the multivariate logistic analysis found that AAA diameter and hs-CRP were independently positively associated with AAA rupture, and HDL-C level was adversely associated with AAA rupture.

**Conclusions:** Our data suggests that larger AAA diameter and higher hs-CRP level are associated with a higher risk of AAA rupture, and higher HDL-C level is associated with a lower risk of AAA rupture. The results of this study may be helpful for the management of AAA patients in southern China.

## INTRODUCTION

Abdominal aortic aneurysm (AAA) is one of the most severe and complex cardiovascular diseases (CVD), causing about 175,000 deaths worldwide per year and affecting 3–9% of

the population over age 65 (*Golledge, 2019*; *Ali et al., 2016*). AAA is defined as abdominal aortic dilation to a diameter of more than 30 mm. Most AAAs are asymptomatic and many patients only develop abdominal symptoms when the AAA is complicated by impending rupture. Once an AAA ruptures, the fatality rate reaches as high as ~90% (*Schanzer & Oderich, 2021*). AAA diameter has been well established as a crucial determinant and independent predictor of rupture (*Brown, Zelt & Sobolev, 2003*), and an AAA diameter size of 55 mm is generally accepted as an indicator of a large AAA in need of endovascular aneurysm repair or open surgery (*Wanhainen et al., 2016*). However, some studies have found that rupture can occur in smaller aneurysms (*Limet, Sakalihassan & Albert, 1991*; *Fillinger et al., 2003*). Therefore, it is important to characterize the risk factors associated with AAA rupture to decrease its incidence.

The natural progression of AAAs has been well described and shows an initial linear progression, followed by exponential growth until rupture (*Golledge, 2019*). Though the exact mechanisms are not yet fully known, some risk factors have been found to influence this process. Since most AAAs are found in individuals with advanced atherosclerosis, it is presumed that atherosclerosis contributes to AAA formation by decreasing the elastic recoil of the aortic wall (*Toghill, Saratzis & Bown, 2017*). AAA and atherosclerotic diseases also share some risk factors, like aging, being male, smoking, hyperglycemia, and hypertension (*Aggarwal et al., 2011*). In addition, genetic factors play an important role in AAA development, with heritability estimated to be as high as 70% (*Wahlgren et al., 2010*). However, the knowledge of risk factors associated with AAA rupture is relatively limited. The most relevant factor for rupture is aneurysm diameter (*Kuivaniemi et al., 2015*). The risk of rupture increases in proportion to diameter (*Chaikof et al., 2018*). Some studies suggest that smoking is a risk factor for both AAA development and rupture, as smokers have a higher annual aneurysmal growth rate than non-smokers or ex-smokers (*Sweeting et al., 2012*). Surprisingly, diabetes has been found to be negatively associated with AAA rupture despite being an important risk factor of atherosclerosis (*Takagi & Umemoto, 2016*). Though the majority of AAAs occur in males, AAAs in female patients are more likely to rupture (*Khan et al., 2015*). However, most of these findings are from studies in western populations, and there are very few studies investigating AAAs in the southern Chinese population.

The present study compares clinical characteristics and laboratory parameters between rupture and non-rupture AAA patients. This study aims to identify risk factors associated with AAA rupture.

## METHODS

### Patients

All AAA patients included in this retrospective study were hospitalized at the Meizhou People's Hospital between January 2018 and December 2022 and were diagnosed based on a computed tomography angiography (CTA) scan. Patients who suffered severe symptomatic abdominal pain were screened with abdominal ultrasound; if the ultrasound suggested an AAA, a CTA scan was subsequently arranged to confirm the diagnosis. AAA cases were diagnosed based on the clinical practice guidelines of the European Society for

Vascular Surgery (ESVS) (*Wanhainen et al., 2019*). An aortic aneurysm with a maximal diameter of more than 3 cm was identified as an AAA. Patients who were also diagnosed with abdominal aorta pseudoaneurysm, aortic dissecting aneurysm, perforating ulcer of abdominal aorta or advanced cancers were excluded (*Qiu et al., 2022*). AAA patients who lacked preoperative CTA or important clinical data, such as risk factors or laboratory results, were also excluded.

All diagnoses from CTA findings were reviewed by two cardiologists. The first cardiologist wrote the CTA report of the characteristics of the AAA, and the second cardiologist confirmed the diagnosis. AAAs were only diagnosed if both cardiologists agreed. AAAs were divided into three categories based on location: ① infrarenal AAAs are located in or above the renal artery; ② pararenal AAAs are located within the 15mm below the renal artery; ③ ; suprarenal AAAs are located more than 15 mm below the renal artery. AAA cases were defined as ruptured if a retroperitoneal and/or intraperitoneal hematoma neighboring the aneurysm sac was observed in the preoperative CTA. This study was conducted in accordance with the declaration of Helsinki. The protocol of the study was approved by the Ethics Committee of the Meizhou People's Hospital (Huangtang Hospital), Meizhou Academy of Medical Sciences (MPH-HEC 2021-C-113). For informed consent, patients were contacted by telephone, and verbal consent was obtained from each patient before the study began.

## Data collection

Demographic information and clinical characteristics including age, gender, smoking history, alcohol drinking habits, history of hypertension, history of diabetes, history of CHD, laboratory test data, and CTA data were all collected from the hospital electronic medical record system.

Patients were defined as alcohol drinkers if they consumed more than 30 g/day of alcohol. Patients were diagnosed with hypertension either by having blood pressure ≥130/85 mmHg or regularly taking antihypertensive medication. Diabetes mellitus was diagnosed by either a fasting glucose of ≥126 mg/dL or regular antidiabetic treatment. Patients were diagnosed with dyslipidemia if they met any of the following conditions: (i) serum TC ≥ 5.17 mmol/L, (ii) serum TG ≥ 1.7 mmol/L, (iii) serum HDL-C < 1.04 mmol/L, (iv) serum LDL-C ≥ 4.14 mmol/L, (v) or were taking antidyslipidemic medication.

Fasting blood samples were collected the morning after admission and before AAA treatment. Routine laboratory analyses, including HbA1c, LVEF, cTnI, BNP, hsCRP, creatinine, D-dimer, and serum lipid profiles, were performed immediately upon admission. If lab results, such as cTnI, were acquired multiple times during the hospital stay, the first results, tested prior to AAA treatment, were included in the analysis.

For patients with multiple scans during the study period, pre-operative CTA data were used to analyze the diameter and location of AAAs in this study.

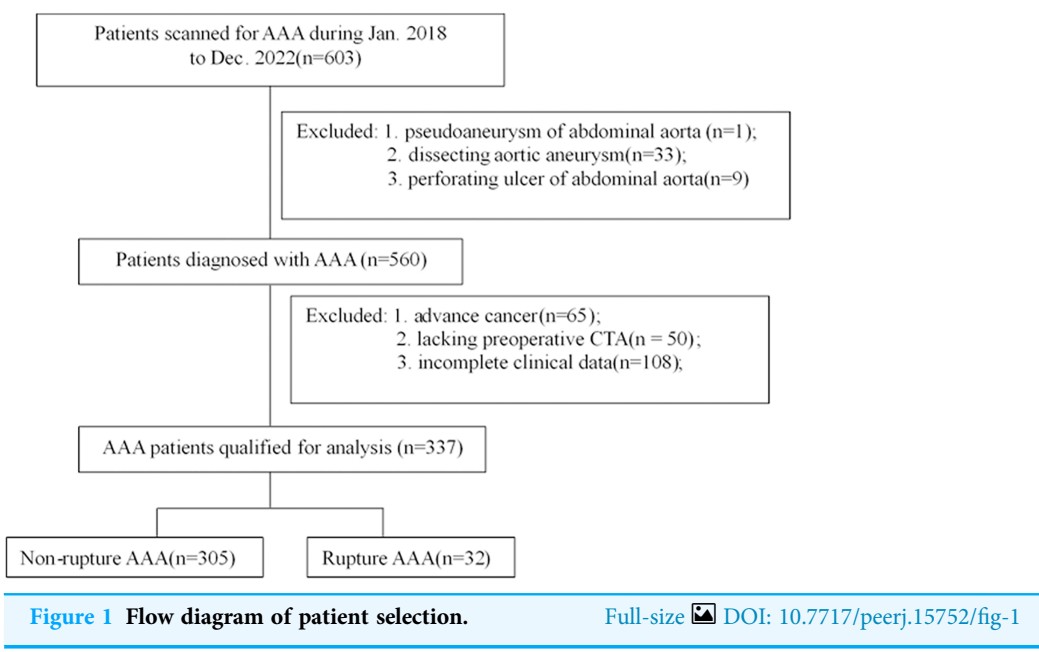

**Figure 1 Flow diagram of patient selection.**

## Data analysis

The statistical analysis was performed using SPSS software version 22.0 (IBM Corp., Armonk, NY, USA). Continuous variables were expressed as means ± standard deviation, and categorical variables were expressed as absolute numbers (percentages).

The Kolmogorov–Smirnov test was applied to determine the normality of the distribution of continuous variables. Continuous variables were compared using either Student's t tests between two groups, or one-way ANOVA between three groups. Categorical variables were compared using the chi-square ($\chi^2$) test. The logistic regression analysis was used to identify independent risk factors for rupture. Odds ratios (ORs) were calculated by adjusting for variables, such as age and gender. All tests were two-sided, and a $P$ value <0.05 was considered statistically significant.

## RESULTS

### Patient characteristics

During the study period, a total of 603 patients who suffered symptomatic severe abdominal pain were scanned with abdominal ultrasound for suspected AAA. After selection according to the inclusion and exclusion criteria, 337 patients diagnosed with AAA were included for analysis (Fig. 1). AAA rupture occurred in 32 of the 337 (9.5%) patients. The clinical characteristics of the included AAA patients are presented in Table 1. The average age of the patients was 73.19 ± 10.26, and 85.76% of the patients were male. There were no significant differences in characteristics between rupture and non-rupture patients.
**Table 1 Clinical characteristics of AAA patients included in the study.**

| Variables | Total (*n* = 337) | Rupture AAA (*n* = 32) | Non-rupture AAA (*n* = 305) | *P* |
|---|---|---|---|---|
| Age, year | 73.19 ± 10.26 | 71.56 ± 10.24 | 73.36 ± 10.26 | 0.347 |
| Male, *n* (%) | 289 (85.76%) | 26 (81.2%) | 263 (86.2%) | 0.443 |
| Smoking state, *n* (%) | 95 (28.19%) | 5 (15.6%) | 90 (29.5%) | 0.097 |
| Drinking state, *n* (%) | 20 (5.93%) | 2 (6.2%) | 18 (5.9%) | 1.000 |
| Hypertension, *n* (%) | 201 (59.64%) | 17 (53.1%) | 184 (60.3%) | 0.429 |
| Diabetes mellitus, *n* (%) | 51 (15.13%) | 8 (25.0%) | 43 (14.1%) | 0.118 |
| Dyslipidemia, *n* (%) | 183 (54.3%) | 21 (65.6%) | 162 (53.1%) | 0.177 |
| CHD history, *n* (%) | 152 (45.10%) | 12 (37.5%) | 140 (45.9%) | 0.364 |

Note:
CHD, coronary heart disease.

**Table 2 Comparison of clinical features between rupture and non-rupture AAA patients.**

| Variables | Rupture AAA (*n* = 32) | Non-rupture AAA (*n* = 305) | *P* |
|---|---|---|---|
| Diameter (cm) | 6.58 ± 2.04 | 4.63 ± 1.61 | <0.001 |
| HbA1c (%) | 6.00 ± 0.61 | 6.08 ± 0.90 | 0.672 |
| LVEF (%) | 61.10 ± 6.71 | 59.01 ± 10.29 | 0.130 |
| cTnI (ng/ml) | 0.824 ± 3.50 | 1.23 ± 5.28 | 0.683 |
| BNP (pg/ml) | 460.69 ± 1,064.06 | 354.85 ± 823.96 | 0.515 |
| hsCRP (mg/ml) | 91.11 ± 71.42 | 46.66 ± 62.28 | 0.001 |
| Creatinine (mg/dL) | 161.02 ± 79.34 | 117.42 ± 110.29 | 0.033 |
| D-dimer (mg/L) | 5.63 ± 6.02 | 3.81 ± 4.46 | 0.149 |
| HDL-c (mmol/L) | 0.88 ± 0.25 | 1.14 ± 0.36 | <0.001 |
| LDL-c (mmol/L) | 1.99 ± 0.70 | 2.51 ± 0.94 | 0.005 |
| TG (mmol/L) | 1.62 ± 1.10 | 1.55 ± 1.32 | 0.776 |
| TC (mmol/L) | 3.67 ± 1.05 | 4.43 ± 1.25 | 0.002 |
| ApoA1 (g/L) | 0.77 ± 0.17 | 0.97 ± 0.23 | <0.001 |
| ApoB1 (g/L) | 0.67 ± 0.20 | 0.80 ± 0.25 | 0.009 |
| ApoB/ApoA1 | 0.90 ± 0.30 | 0.87 ± 0.50 | 0.761 |

Note:
LVEF, left ventricular ejection fraction; cTnI, cardiac troponin I; BNP, brain natriuretic peptide; hsCRP, high sensitive C reaction protein; HCY, homocysteine; HDL-C, high-density lipoprotein cholesterol; LDL-C, low-density lipoprotein cholesterol; TG, triglycerides; TC, total cholesterol; ApoA1, apoliprotein A1; ApoB, apoliprotein B.

## Comparison of clinical features between rupture and non-rupture AAA patients

We compared the clinical features between rupture and non-rupture AAA patients. As shown in Table 2, the AAA diameter in the rupture group was significantly larger than that in the non-rupture group (6.58 ± 2.04 *vs.* 4.63 ± 1.61, *P* < 0.001). Compared to non-rupture patients, rupture patients had higher levels of hs-CRP (91.11 ± 71.42 *vs.* 42.78 ± 61.01, *P* < 0.001) and creatinine (161.02 ± 79.34 *vs.* 117.42 ± 110.29, *P* = 0.033). AAA rupture patients exhibited lower levels of HDL-C, LDL-C and TC than non-rupture patients (0.88 ± 0.25 *vs.* 1.14 ± 0.36, *P* < 0.001; 1.99 ± 0.70 *vs.* 2.51 ± 0.94, *P* = 0.005; 3.67 ± 1.05 *vs.* 4.43 ± 1.25, *P* < 0.002; respectively). Additionally, ApoA1 and ApoB levels were

**Table 3 Comparison of clinical features between patients with AAAs of different diameters.**

| Variables | Diameter <5.5 cm (n = 230) | Diameter ≥5.5 cm (n = 107) | P |
|---|---|---|---|
| Rupture, n (%) | 9 (3.91%) | 23 (21.50%) | <0.001 |
| Age, year | 73.71 ± 10.47 | 72.05 ± 9.75 | 0.154 |
| HbA1c (%) | 6.09 ± 0.96 | 6.03 ± 0.71 | 0.603 |
| LVEF (%) | 59.00 ± 10.30 | 59.67 ± 9.39 | 0.585 |
| cTnI (ng/ml) | 1.26 ± 5.20 | 1.01 ± 4.92 | 0.709 |
| BNP (pg/ml) | 355.73 ± 792.94 | 392.29 ± 983.29 | 0.744 |
| hsCRP (mg/ml) | 52.62 ± 66.07 | 69.60 ± 62.26 | 0.745 |
| Creatinine (mg/dL) | 118.24 ± 114.24 | 126.11 ± 110.04 | 0.896 |
| D-dimer (mg/L) | 3.48 ± 4.15 | 5.04 ± 5.44 | 0.027 |
| HDL-C (mmol/L) | 1.12 ± 0.34 | 1.11 ± 0.40 | 0.905 |
| LDL-C (mmol/L) | 2.49 ± 0.97 | 2.40 ± 0.85 | 0.392 |
| TG (mmol/L) | 1.56 ± 1.36 | 1.54± 1.18 | 0.871 |
| TC (mmol/L) | 4.35 ± 1.23 | 4.40 ± 1.29 | 0.757 |
| ApoA1 (g/L) | 0.95 ± 0.24 | 0.95 ± 0.20 | 0.895 |
| ApoB (g/L) | 0.78 ± 0.25 | 0.78 ± 0.24 | 0.991 |
| ApoB/ApoA1 | 0.89 ± 0.55 | 0.85 ± 0.27 | 0.427 |

Note:
SBP, systolic blood pressure; DBP, diastolic blood pressure; HbA1c, glycosylated hemoglobin, Type A1C; LVEF, left ventricular ejection fractions; cTnI, cardiac troponin I; BNP, brain natriuretic peptide; hsCRP, high sensitive C reaction protein; HCY, homocysteine; HDL-C, high-density lipoprotein cholesterol; LDL-C, low-density lipoprotein cholesterol; TG, triglycerides; TC, total cholesterol; ApoA1, apoliprotein A1; ApoB, apoliprotein B.

lower in AAA rupture patients than in non-rupture patients (0.77 ± 0.17 *vs.* 0.97 ± 0.23, $P < 0.001$; 0.67 ± 0.20 *vs.* 0.80 ± 0.25, $P = 0.009$). There were no significant differences between rupture and non-rupture patients for common cardiovascular markers, such as cTnI and BNP.

## Comparison of clinical features based on AAA size and location, and patient gender

Next, we compared clinical features of AAA patients based on AAA size and location, and patient gender. Aneurysm diameter is the single most important predictor of AAA rupture and is the most common measurement used for AAA management. As shown in Table 3, patients with AAA diameters greater than 5.5 cm had significantly higher rates of rupture (21.50% *vs.* 3.91%, $P < 0.001$). There were no significant differences in clinical features of AAA patients by aneurism diameter, except for D-dimer, which was higher in patients with AAA diameters greater than 5.5cm (5.04 ± 5.44 *vs.* 3.48 ± 4.15, $P = 0.027$). As shown in Table 4, female AAA patients had a higher rupture rate than male patients, but this difference did not reach statistical difference (12.5% *vs.* 9.0%, $P = 0.428$); compared to male patients, female patients were older (76.21 ± 11.42 *vs.* 72.69 ± 9.99, $P = 0.027$) and had higher levels of HDL-C (1.26 ± 0.44 *vs.* 1.10 ± 0.34, $P = 0.020$), TC (4.91 ± 1.52 *vs.* 4.27 ± 1.17, $P = 0.001$) and ApoA1 (1.06 ± 0.27 *vs.* 0.93 ± 0.22, $P = 0.004$). AAAs were then divided into infrarenal, pararenal and suprarenal based on their locations. As shown in Table 5, most AAAs were infrarenal, followed by pararenal and suprarenal. The rupture

**Table 4 Comparison of clinical features between male and female AAA patients.**

| Variables | Male (*n* = 289) | Female (*n* = 48) | *P* |
|---|---|---|---|
| Rupture, *n* (%) | 26 (9.0%) | 6 (12.5%) | 0.428 |
| Age, year | 72.69 ± 9.99 | 76.21 ± 11.42 | 0.027 |
| Diameter (cm) | 4.81 ± 1.72 | 4.83±1.90 | 0.948 |
| HbA1c (%) | 6.09 ± 0.90 | 5.96 ± 0.69 | 0.400 |
| LVEF (%) | 59.11 ± 10.07 | 59.93 ± 9.68 | 0.617 |
| cTnI (ng/ml) | 1.02 ± 4.74 | 2.30 ± 7.10 | 0.309 |
| BNP (pg/ml) | 334.76 ± 781.04 | 581.56 ± 1,223.93 | 0.247 |
| hsCRP (mg/ml) | 53.81 ± 66.33 | 37.72 ± 52.18 | 0.206 |
| Creatinine (mg/dL) | 122.23 ± 96.73 | 117.55 ± 162.79 | 0.785 |
| D-dimer (mg/L) | 3.81 ± 4.49 | 5.46 ± 5.75 | 0.149 |
| HDL-C (mmol/L) | 1.10 ± 0.34 | 1.26 ± 0.44 | 0.020 |
| LDL-C (mmol/L) | 2.44 ± 0.94 | 2.57 ± 0.87 | 0.379 |
| TG (mmol/L) | 1.52 ± 1.23 | 1.79± 1.66 | 0.196 |
| TC (mmol/L) | 4.27 ± 1.17 | 4.91 ± 1.52 | 0.001 |
| ApoA1 (g/L) | 0.93 ± 0.22 | 1.06 ± 0.27 | 0.004 |
| ApoB (g/L) | 0.78 ± 0.25 | 0.81 ± 0.23 | 0.460 |
| ApoB/ApoA1 | 0.89 ± 0.51 | 0.80 ± 0.27 | 0.234 |

rate for infrarenal, pararenal and suprarenal AAAs were 9.84%, 13.64% and 4.55%, respectively. The diameter of pararenal AAAs were significantly larger than infrarenal and suprarenal AAAs ($P < 0.001$). There were no other significant differences in clinical features by aneurysm location.

## Independent risk factors associated with AAA rupture

A binary logistic regression analysis was used to evaluate the clinical features associated with AAA rupture. Based on the results of the previous analysis, gender, smoking history, AAA diameter, HDL-C level, creatinine level and hs-CRP level were input into the regression model. As shown in Table 6, the univariate analysis found that AAA diameter, and creatinine and hs-CRP levels were positively associated with AAA rupture, while HDL-C level was adversely associated with AAA rupture. The multivariate model suggests that AAA diameter (OR = 2.499, 95% CI [1.704 –3.665], $P < 0.001$) and hs-CRP level (OR = 1.017, 95% CI [1.008–1.026], $P < 0.001$) are associated with increased risk of rupture, while higher HDL-C level is associated with decreased risk of rupture (OR = 0.024, 95% CI [0.002–0.243], $P = 0.002$).

## DISCUSSION

In this study, we analyzed the clinical features of hospitalized AAA patients over the last 5 years. Our data suggests that larger AAA diameter and higher hs-CRP level are associated with increased rupture risk, and higher HDL-C level is associated with decreased rupture risk.

**Table 5 Comparison of clinical features between patients by AAA location.**

| Variables | Infrarenal ($n$ = 244) | Pararenal ($n$ = 44) | Suprarenal ($n$ = 22) | $P$ |
|---|---|---|---|---|
| Rupture, $n$ (%) | 24 (9.84%) | 6 (13.64%) | 1 (4.55%) | 0.501 |
| Age, year | 72.02 ± 10.47 | 72.59 ± 8.75 | 71.23 ± 11.56 | 0.728 |
| Diameter (cm) | 4.63 ± 1.72 | 5.84 ± 1.68 | 4.45 ± 1.49 | <0.001 |
| HbA1c (%) | 6.06 ± 0.87 | 6.10 ± 0.50 | 6.09 ± 1.27 | 0.969 |
| LVEF (%) | 59.84 ± 9.53 | 60.08 ± 8.38 | 62.14 ± 6.66 | 0.537 |
| cTnI (ng/ml) | 0.90 ± 4.60 | 0.40 ± 1.44 | 0.02 ± 0.04 | 0.644 |
| BNP (pg/ml) | 267.89 ± 680.62 | 413.03 ± 999.18 | 581.56 ± 1,223.93 | 0.441 |
| hsCRP (mg/ml) | 50.54 ± 62.00 | 38.87 ± 33.76 | 38.52 ± 48.43 | 0.596 |
| Creatinine (mg/dL) | 111.95 ± 79.74 | 130.98 ± 114.95 | 96.28 ± 37.09 | 0.236 |
| D-dimer (mg/L) | 3.87 ± 4.57 | 4.94 ± 5.72 | 3.61 ± 3.27 | 0.450 |
| HDL-C (mmol/L) | 1.13 ± 0.36 | 1.06 ± 0.28 | 1.09 ± 0.36 | 0.448 |
| LDL-C (mmol/L) | 2.47 ± 0.96 | 2.51 ± 0.86 | 2.37 ± 0.80 | 0.850 |
| TG (mmol/L) | 1.56 ± 1.43 | 1.62 ± 0.84 | 1.42 ± 0.74 | 0.842 |
| TC (mmol/L) | 4.35 ± 1.28 | 4.52 ± 1.18 | 4.33 ± 1.04 | 0.716 |
| ApoA1 (g/L) | 0.96 ± 0.22 | 0.94 ± 0.21 | 0.95 ± 0.26 | 0.905 |
| ApoB (g/L) | 0.78 ± 0.25 | 0.82 ± 0.23 | 0.78 ± 0.20 | 0.493 |
| ApoB/ApoA1 | 0.86 ± 0.51 | 0.90 ± 0.28 | 0.91 ± 0.49 | 0.808 |

**Table 6 Independent risk factors associated with AAA rupture.**

| Variables | Univariate | | Multivariate | |
|---|---|---|---|---|
| | OR (95% CI) | $P$ value | Adjusted OR (95% CI) | $P$ value |
| Male | 1.445 [0.561–3.720] | 0.445 | 3.059 [0.679–13.790] | 0.146 |
| Smoking | 2.260 [0.844–6.056] | 0.105 | 4.508 [0.857–23.697] | 0.075 |
| Diameter | 1.811 [1.460–2.247] | <0.001 | 2.499 [1.704–3.665] | <0.001 |
| HDL-C | 0.051 [0.011–0.231] | <0.001 | 0.024 [0.002–0.243] | 0.002 |
| HsCRP | 1.009 [1.003–1.014] | 0.001 | 1.011 [1.005–1.016] | <0.001 |
| Creatinine | 1.002 [1.000–1.005] | 0.048 | 1.002 [1.000–1.005] | 0.063 |

**Note:**
HsCRP, high sensitive C reaction protein; HDL-C, high-density lipoprotein cholesterol.

AAA rupture is the main complication experienced by AAA patients and is a significant cause of mortality (*Sampson et al., 2014*). Rupture occurs when the hemodynamic forces from pulsatile blood flow exceed the strength of the AAA wall. The exact mechanisms of AAA rupture are unknown. There was an observed AAA rupture rate of 9.5% in our study, which is similar to those reported in other population studies (*Lilja, Wanhainen & Mani, 2017*; *Nordon et al., 2011*). Intensive efforts have been made to investigate the risk factors of AAA rupture. AAA diameter is currently the most relevant predictor of AAA rupture. The risk of rupture increases from a risk of less than 5% per year for AAAs of 4.0–4.9 cm to a risk of more than 30% for AAAs of >7.0 cm (*Chaikof et al., 2009*). Clinical guidelines recommend that small asymptomatic AAAs (<5.0 cm in women and <5.5 cm in men) be

managed by interval surveillance imaging, and large AAAs be considered for surgical repair (*Modarai, 2019*). In our study, AAA diameter was significantly larger in rupture patients than in non-rupture patients, and patients with an AAA diameter ≥5.5 cm had a much higher rupture rate than those with an AAA diameter <5.5 cm. The multivariate logistic regression model also supported the result that larger AAA diameter is strongly associated with increased rupture risk.

The prevalence of AAA is much lower in women, but female AAA patients face higher risks of rupture. A large meta-analysis by *Sweeting et al. (2012)* found that the rupture of small AAAs under surveillance is about four times more likely to occur in females than males. Healthy males typically have a much larger aortic diameter compared to females (*Norman & Powell, 2007*), which explains the higher risk of rupture in women compared to men with similar size AAAs. In our study cohort, 85.76% of the AAA patients were male, and the rupture rate was higher in female patients by a small margin, but did not reach statistical significance. Evidence suggests that smoking is a risk factor for AAA formation (*Pujades-Rodriguez et al., 2015*). Observational studies show that quitting smoking decreases the risk of AAA formation and growth (*Sweeting et al., 2012*; *Tang et al., 2016*). However, our study did not observe a significant correlation between smoking and AAA rupture. This may be explained, in part, by the fact that smoking history was self-reported in our study, and patients may not have been entirely truthful.

The role of serum lipid profile is well defined in cardiovascular diseases, and LDL-C lowering therapies are known to reduce CHD risk. In AAA, however, the conclusions of research on the role of serum lipid profile in AAA remain inconsistent. Landlad et al. investigated the risk factors associated with AAA development and found that triglycerides were significantly elevated in AAA patients (*Lindblad, Borner & Gottsater, 2005*). Meta-analyses of observational studies do show a consistent inverse association of HDL-C with AAA risk (*Takagi, Manabe & Umemoto, 2010*). *Harrison et al. (2018)* reported that lipids play an important role in the etiology of AAA, suggesting that LDL-C lowering therapies may be an effective treatment for AAA. *Delbosc et al. (2013)* reported that impaired HDL antioxidant capacity was found in AAA patients, highlighting the significant role of HDL in AAA development. Serum triglycerides may also be a strong risk factor for AAA rupture (*Watt et al., 1998*). In the present study, serum lipids of rupture patients, including HDL-C, LDL-C and TC were lower than in non-rupture patients. The multiple regression model also suggested that higher HDL-C level is associated with reduced AAA rupture.

Chronic aortic inflammation is believed to lead to destruction of the aortic media and to vascular smooth muscle cell apoptosis and dysfunction as a result of the release of a range of proteolytic enzymes, such as matrix metalloproteinases and cysteine proteases, oxidation-derived free radicals, cytokines and related products (*Fillinger et al., 2003*). Thus, inflammatory markers may also be associated with AAA rupture. A previous study found that a range of cytokines, particularly IL6, are present at higher concentrations in the plasma of patients with AAA (*Golledge et al., 2008*). Our data showed that AAA rupture patients had higher levels of hs-CRP and serum creatinine. After adjusting for other factors, hs-CRP presented as a modest risk factor for AAA rupture.

This study had some limitations. Though this retrospective study included 5 years of AAA patients, the sample size was not large because of the low incidence rate of AAA. This was also a retrospective, single-center study. Therefore, the results of this study only suggest an association between clinical factors and AAA rupture, but cannot establish causality. Until further research is performed, caution should be applied when considering the specificity of these candidate biomarkers for AAA.

## CONCLUSION

The present study investigated the clinical risk factors associated with AAA rupture in patients from southern China. Our data suggest that larger AAA diameter and higher hs-CRP level are associated with increased risk of rupture, while higher HDL-C level is associated with reduced risk of AAA rupture.

### Funding

This research was supported by the National Natural Science Foundation for Young Scientists of China (82000410); Guangdong Natural Science Foundation (2022A1515011860); Medical Research Foundation of Guangdong Province (A2023324); Scientific Research and Cultivation Project of Meizhou People's Hospital (PY-C2021045, PY-C2022017). The funders had no role in study design, data collection and analysis, decision to publish, or preparation of the manuscript.

### Grant Disclosures

The following grant information was disclosed by the authors:
National Natural Science Foundation for Young Scientists of China: 82000410.
Guangdong Natural Science Foundation: 2022A1515011860.
Medical Research Foundation of Guangdong Province: A2023324.
Scientific Research and Cultivation Project of Meizhou People's Hospital: PY-C2021045, PY-C2022017.

### Competing Interests

The authors declare that they have no competing interests.

### Author Contributions

- Sudong Liu performed the experiments, authored or reviewed drafts of the article, and approved the final draft.
- Caifu Long analyzed the data, prepared figures and/or tables, and approved the final draft.
- Yuanjia Hong analyzed the data, prepared figures and/or tables, and approved the final draft.

- Xiaodong Gu performed the experiments, authored or reviewed drafts of the article, and approved the final draft.
- Ruiqiang Weng performed the experiments, authored or reviewed drafts of the article, and approved the final draft.
- Zhixiong Zhong conceived and designed the experiments, authored or reviewed drafts of the article, and approved the final draft.

## Human Ethics

The following information was supplied relating to ethical approvals (*i.e.*, approving body and any reference numbers):

This protocol of the study had been approved by the Ethics Committee of the Meizhou People's Hospital (Huangtang Hospital), Meizhou Academy of Medical Sciences (MPH-HEC 2021-C-113).

## Data Availability

The raw measurements are available in the Supplemental File.

## Supplemental Information

Supplemental information for this article can be found online at http://dx.doi.org/10.7717/peerj.15752#supplemental-information.

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
