# Peer review of "Prevalence of risk factors associated with rupture of abdominal aortic aneurysm (AAA): a single center retrospective study"

_PeerJ, doi:10.7717/peerj.15752_

## Round 0.1 · original submission · Major Revisions

I sincerely thank the authors for submitting their manuscript for consideration. I commend the authors for taking up this important work. The topic is relevant and timely. I do not see any ethical issues. Data collection appropriate. However, significant methodology flaws exist in the data analysis and interpretation. The reviewers have raised significant and relevant criticisms. Hence, we are unable to accept the paper in its current form. My specific comments as follows:

1) The study is a cross-sectional observational study. The authors report that fasting glucose and total cholesterol are risk factors for AAA rupture whereas HbA1C is not a risk factor. There is likely no biological explanation to these findings. These unusual associations are likely due to the phenomenon of ‘reverse causation’ that is an inherent limitation of cross-sectional studies. A patient with AAA rupture might have not been eating well for the last 3-4 days prior to hospitalization and hence fasting glucose might be low and hence the data shows that fasting glucose was normal in patients with AAA rupture while HbA1C was high. This reasoning is true with BP also. Hence, reverse causation is biasing this study and I do not believe there is a way to adjust this.
I would recommend a simpler approach to the paper as a whole. Instead of trying to identify risk factors, a study question that is not appropriate to the study design the authors have chosen, just keep the paper as a prevalence paper. In other words, keep it simple and report prevalence of risk factors in a single center of patients presenting with AAA. This will be a more valid and appropriate approach for a cross-sectional study.

2) I would avoid using parameters like fasting glucose and presenting blood pressure as variables because they change rapidly and hence are not relevant for assessing associations with a chronic disease like AAA.

3) Introduction and discussion should be significantly reduced and tailored to the topic of interest. They meander a lot away from the topic of interest.

4) Significant language editing, typo and grammar corrections are required prior to submission to make the paper acceptable

I do believe that extensive revisions are needed based on the reviewers' and my comments. I sincerely request the authors to revise and re-submit for consideration.

Reviewer 1 ·

Basic reporting

The authors were trying to identify modifiable risk factors of AAA rupture in Southern China in order to better risk stratify AAA and employ early preventive measurement. They did an elegant job, however further modification is expected before it is accepted.

Experimental design

Author discussed AAA based on the size, what about gender difference and AAA location?

gender seems to play a different role in AAA development. Male have higher risks to develop AAA, while female have higher risks of AAA rupture. Any risk factor differences between these two groups?

Any influence of the AAA location? Infrarenal vs Juxtarenal vs Pararenal vs Suprarenal (visceral)?

Validity of the findings

very interesting and confusing finding

1. high fasting glucose level is associated higher risk of rupture, but overall the higher A1C is not associated with the rupture risk, any hypothesis?

2. lower total cholesterol is linked to lower risk of rupture, but lower level of HLD and LDL did not demonstrated any protective effects. What about VLDL? Any hypothesis?

3. higher HsCRP is the result of AAA rupture or the cause of AAA rupture?

Additional comments

1. Would not complain too much about the English writing, but it could be improved
2. the title is misleading, the whole manuscript discussed the risk of rupture, rather than AAA development. Or the authors meant to say progression rather than development?
3. Please spell out all abbreviation when it is used the first time in the manuscript.
4. Please specify in conclusion that larger diameter, higher fasting glucose are risk factors of rupture, while higher total cholesterol is protective against rupture. You need a clear conclusion so that the audience gets your take home message immediately
5. Assume your hospital in southern China, is it a national hospital drawing patients nationally or a regional hospital serving patients locally? Any differences between your data to the national date from China?

Reviewer 2 ·

Basic reporting

Dear Editor,

PeerJ.

Subject: Peer review

I enjoyed reviewing the manuscript submitted to your journal, entitled ‘Risk factors associated with development and rupture of abdominal aortic aneurysm (AAA) in southern China.’

The authors undertook a retrospective study to assess the risk factors for the development and rupture of AAA in the southern Chinese population. This single-center study analyzed data over 5 years and utilized logistic regression analysis to evaluate the risk factors associated with AAA rupture. I commend the authors for conducting this study, which provides valuable insights into identifying risk factors in the local population.

The review has major concerns as below.

Major points:
1. The development and rupture of abdominal aortic aneurysms (AAA) are well-studied phenomena with several established risk factors. These include smoking, male sex, advancing age, and family history of the risk of developing AAA. Risk factors for AAA rupture include a large diameter greater than 5.5 cm, elevated blood pressure, female sex, smoking, and an AAA expansion rate of greater than 0.5 cm per year. In this study, the authors found that systolic blood pressure was not associated with rupture, which is counterintuitive given the established link between blood pressure and AAA development. However, they did find a new risk factor for AAA rupture: fasting glucose. Notably, the study did not include the effects of smoking, gender, or AAA rate of expansion in the multivariate analysis, which may affect the accuracy of the results. Encourage authors to include these variables.
2. The dataset contains 32 events, which is a relatively small sample size. A general rule of thumb in statistical analysis is to include no more than 10 variables per event to maintain statistical power. Based on this rule, only three variables should be included in a regression analysis to maintain the validity of the results. However, if more than three risk factors are studied, the small sample size may pose a significant threat to the validity of the analysis. Therefore, it is important to carefully select the variables for analysis and interpret the results with caution, given the limited number of events in the dataset.
3. The authors' assertion that verbal consent was obtained from each patient before the study is inconsistent with the intended retrospective design of the study, presenting a contradiction that may require further clarification.
4. Given that Table 2 does not appear to align with the intended scope or focus of the study, it may be advisable to consider including it as a supplementary rather than a core component of the analysis.
5. The topics of peak wall stress and PWRI are beyond the scope of the present paper and may therefore be excluded from the current discussion.
6. The authors' suggestion that total cholesterol had a protective effect may be inappropriate, as the study was not designed to establish causation but rather to explore potential associations through logistic regression. As such, it may be more accurate to describe any observed relationships as correlations rather than causal links.

Minor points:
1. The English should be improved to ensure that an international audience understands the text. I suggest you have a colleague proficient in English and familiar with the subject matter review your manuscript or contact a professional editing service.
2. It would benefit the authors to maintain a more focused approach rather than straying off-topic. In particular, the introduction and discussion sections may benefit from being streamlined to convey their intended message better.

Experimental design

See above

Validity of the findings

See above

Additional comments

See above

Reviewer 3 ·

Basic reporting

- Overall, the article seems to be easy to understand and clear. There are a few typos (E.g. line 172 – “fast glucose”) in the manuscript that will likely need editing.
- Figures are easy to understand, and tables are helpful.

Experimental design

- The research question states “to characterize the clinical risk factors that were associated with the progression and rupture of AAA, and make a contribution to the management of local AAA patients.” In my opinion, this needs to be significantly modified. I do not think the study methodology is capable of answering this question. It doesn't study progression at all. In my opinion, the study is more an exploration of possible factors associated with AAA rupture.
- The authors share that this was a retrospective study, and all AAA cases from 01/2018 to 12/2022 were included based on CTA scan results (lines – 92-94). Figure 1 states 01/2018 to 10/2022. One of these should be corrected.
- It will be helpful to provide more details on the inclusion criteria - how were the AAA cases identified? Were the CTA scans done only to evaluate for abdominal aorta associated pathology? Were these patients coming in for symptomatic severe abdominal pain that received a CTA? In the US, many patients receive screening with abdominal ultrasound for AAA, and they subsequently get CTA scans annually. Also, in my practice, most of the AAA are diagnosed incidentally, while looking for something else on a routine CT abdomen with contrast – this may not be true in this setting, but wondering how these were handled.
- Figure 1 states 603 patients got scanned for AAA. Were they all unique patients? How were patients with multiple CT scans handled? Did all of them get a pre-operative CTA (second CTA)? The authors share a preoperative CTA was obtained (lines 96-97). This should be clarified. It will be most helpful to share how the data was handled using language such as “XX AAA scans were observed in the study duration, that reflected XX unique patients.“ For patients with multiple scans, it should be important to state which ones were included in the study.
- The authors also state that certain populations were excluded (abdominal aorta pseudoaneurysm, aortic dissecting aneurysm, perforating ulcer of abd aorta, advanced cancers). It will be helpful to add supporting data for this exclusion.
- In line 97-98, authors state that the diagnosis from CTA findings were reviewed by two cardiologists. It will be helpful to add their exact role (e.g. whether they were confirming the diagnosis of CTA, measurements, etc.). If agreement between the cardiologists was evaluated, it will be helpful to add that also.
- The authors state that “verbal consent was obtained from each patient before the study beginning”. (lines 101-102) If the patient or their families were contacted retrospectively, this should be explicitly stated.
- Data collection – the authors state that demographic characteristics and clinical characteristics (lab tests, echo findings, CTA results) during the in-hospital stay were obtained from medical records. It will be helpful to share the timeline about when this information was obtained in relation to imaging and hospital stay. Were these obtained at presentation? In my opinion, the context for each of these tests is really important and it's difficult to interpret these tests when the context is not known. Some of these tests can vary almost daily due to multiple factors (e.g. fasting blood glucose, Cr). If there were multiple data points (e.g. patients with multiple blood glucose levels, Cr) how were these handled? Some of tests (e.g. lipid panel, A1c) would likely not vary much in the inpatient setting. It will be helpful to add more detail to clarify all of this. If this was all part of one hospitalization, it will be helpful to add details about what a typical hospitalization looked like (e.g. approximate length of stay), if that data is available.
- In figure 1, the patients state that those with incomplete clinical data (108, ~20%) were excluded. Will be helpful to add in the text, what parameters were missing.

Validity of the findings

- In the authors' dataset over a 5 year period, ~10% patients experienced a rupture. This is similar to other estimates (~1-5% annually). It may be helpful for the authors to state this and comment on how the populations were similar/different in the discussion section.
- In table 1, the group with the ruptured AAA seems to have more diabetics (~25%) compared to the non-ruptured AAA group. This may explain the differences seen in Table 3 between the groups noted – higher creatinine (in the ruptured AAA group) and lower total cholesterol (likely statin use?). Adding other factors to Table 3 will be helpful – namely SBP, fasting blood glucose, diameter.
- In my opinion, understanding and interpreting the implications/association of fasting blood glucose (which is highly contextual) in the inpatient setting is challenging. Hb A1c would be a better indicator that would highlight long-term glycemic control. Fasting blood glucose likely indicates glycemic control during the hospitalization (~24 hours). It will be helpful to learn whether binary logistic regression was performed with Hb A1c, and what it revealed.
- The authors should reframe their discussion. AAA progression cannot be studied using these methods (see comment above on the research question).
- The authors discuss cholesterol and triglycerides in lines 215-230. This section needs to be modified. It seems based on Table 3, the total cholesterol in rupture AAA group was lower (3.67) compared to non-ruptured AAA group (4.43). This is interesting. This is also seen in Table 4 – higher total cholesterol seems to be protective? (lower odds ratio). This needs to explored further.
- The study has a lot of methodological limitations as highlighted above. Would be helpful to add more details in that section.

---

## Round 0.2 · Minor Revisions

I wholeheartedly congratulate the authors for their efforts in revising this manuscript. The reviewer comments were extensive, and the authors have done a fine job with the revisions. Technically I think the paper is good. This is as best as it gets with the limitations of the data. The analysis is well done.

However, I sincerely request the authors to please take another look at the grammar and English in their manuscript. The validity of the paper will be lost if typos and grammar errors are noted. I request your kind consideration to thoroughly proofread the manuscript for English and grammar. May help to seek help from a language editing service if needed. I sincerely hope the authors will appreciate my efforts and suggestions

Reviewer 1 ·

Basic reporting

The authors were trying to identify modifiable risk factors of AAA rupture in Southern China in order to better risk stratify AAA and employ early preventive measurement. They did an elegant job and the manuscript is ready to be polished after the revision.

Experimental design

the authors addressed my concerns

Validity of the findings

the authors addressed my concerns

Reviewer 2 ·

Basic reporting

Improved

Experimental design

No change

Validity of the findings

No change

Additional comments

Improved manuscript over all

Reviewer 3 ·

Basic reporting

Overall, significant improvements done by the authors. Language still needs improvement due to multiple errors noted throughout the manuscript.

Experimental design

The authors have made significant improvements as suggested. No other comments

Validity of the findings

No comments

Additional comments

Overall, multiple errors in English language and grammar noted throughout the manuscript that need to be rectified.

---

## Round 0.3 · accepted · Accept

I sincerely thank the authors for all their efforts. The paper reads well and I am optimistic it will be a valid addition to the care of patients with AAA in China. Congrats!